**Data Availability Statement:** Marine mammal sighting data from DBO research cruises are available here: arcticdata.io (doi:10.18739/

## RESEARCH ARTICLE

# Changes in gray whale phenology and distribution related to prey variability and ocean biophysics in the northern Bering and eastern Chukchi seas

Sue E. Moore[1]*, Janet T. Clarke[2], Stephen R. Okkonen[3], Jacqueline M. Grebmeier[4], Catherine L. Berchok[5], Kathleen M. Stafford[6¤]

1 Center for Ecosystem Sentinels, University of Washington, Seattle, WA, United States of America, 2 Cooperative Institute for Climate, Ocean and Ecosystem Studies, University of Washington, Seattle, WA, United States of America, 3 Institute of Marine Science, University of Alaska Fairbanks, Fairbanks, Alaska, United States of America, 4 Chesapeake Biological Laboratory, University of Maryland Center for Environmental Science, Solomons, MD, United States of America, 5 Marine Mammal Laboratory, Alaska Fisheries Science Center, NOAA, Seattle, WA, United States of America, 6 Applied Physics Laboratory, University of Washington, Seattle, WA, United States of America

¤ Current address: Marine Mammal Institute, Oregon State University, Newport, OR, United States of America

* moore4@uw.edu

## Abstract

Changes in gray whale (*Eschrichtius robustus*) phenology and distribution are related to observed and hypothesized prey availability, bottom water temperature, salinity, sea ice persistence, integrated water column and sediment chlorophyll *a*, and patterns of wind-driven biophysical forcing in the northern Bering and eastern Chukchi seas. This portion of the Pacific Arctic includes four Distributed Biological Observatory (DBO) sampling regions. In the Bering Strait area, passive acoustic data showed marked declines in gray whale calling activity coincident with unprecedented wintertime sea ice loss there in 2017–2019, although some whales were seen there during DBO cruises in those years. In the northern Bering Sea, sightings during DBO cruises show changes in gray whale distribution coincident with a shrinking field of infaunal amphipods, with a significant decrease in prey abundance (r = -0.314, p<0.05) observed in the DBO 2 region over the 2010–2019 period. In the eastern Chukchi Sea, sightings during broad scale aerial surveys show that gray whale distribution is associated with localized areas of high infaunal crustacean abundance. Although infaunal crustacean prey abundance was unchanged in DBO regions 3, 4 and 5, a mid-decade shift in gray whale distribution corresponded to both: (i) a localized increase in infaunal prey abundance in DBO regions 4 and 5, and (ii) a correlation of whale relative abundance with wind patterns that can influence epi-benthic and pelagic prey availability. Specifically, in the northeastern Chukchi Sea, increased sighting rates (whales/km) associated with an ~110 km (60 nm) offshore shift in distribution was positively correlated with large scale and local wind patterns conducive to increased availability of krill. In the southern Chukchi Sea, gray whale distribution clustered in all years near an amphipod-krill 'hotspot' associated with a 50-60m deep trough. We discuss potential impacts of observed and inferred prey shifts on

A26T0GX06). The ASAMM data are available here: https://www.fisheries.noaa.gov/resource/data/1979-2019-aerial-surveys-arctic-marine-mammals-historical-database PAM Data. Alaska Fisheries Science Center, 2021: AFSC/NMML: Acoustics long-term passive monitoring using moored autonomous recorders in the Bering, Chukchi, and Western Beaufort Seas, 2007-2012, https://www.fisheries.noaa.gov/inport/item/17343. Crustacean abundance and coincident environmental data are available on the DBO project page at the NSF Arctic Data Center archive, here: <https://arcticdata.io/catalog/portals/DBO/Data>. Note: When macrofaunal replicates were only 2-3 grabs/station (2016-2019) due to covid laboratory closures, the data sets will be available on the authors website until full processing of all replicates occurs (https://arctic.cbl.umces.edu). NCEP winds are available here: https://psl.noaa.gov/data/gridded/data.ncep.reanalysis.html. Bering Strait mooring data (including A3 transports) are available here: http://psc.apl.washington.edu/HLD/Bstrait/Data/BeringStraitDownloadregister.html; additional information on Bering Strait moorings here: http://psc.apl.washington.edu/HLD/Bstrait/bstrait.html.

**Funding:** Funding CB - PAM acoustic data: Interagency Agreement between BOEM and AFSC (ARCWEST IA# M12PG00021); grant from the Office of Naval Research, Marine Mammals and Biology Program (Award Number: N000141812792); internal-NOAA grant from the NOAA Fisheries, Office of Science and Technology Ocean Acoustics Program. JC - ASAMM visual data – Interagency Agreements between BOEM and AFSC including M07RG13260, M11PG00033, M16PG00013, and M17PG00031 KS -Marine Mammal DBO Watch visual data: National Science Foundation awards ARC-1107106 ARC-0855828, PLR-1603259, NPRB A94-00, ONR N00014-17-1-2274 JG - National Science Foundation Office of Polar Programs (awards OPP 1204082, 1702456 and 1917469) and National Oceanic and Atmospheric Administration Arctic Research Program (awards CINAR 22309.07 and 25984.02) SM & SO – no funding received for this study The funders had no role in study design, data collection and analysis, decision to publish, or preparation of the manuscript.

**Competing interests:** The authors have declared that no competing interests exist.

gray whale nutrition in the context of an ongoing unusual gray whale mortality event. To conclude, we use the conceptual Arctic Marine Pulses (AMP) model to frame hypotheses that may guide future research on whales in the Pacific Arctic marine ecosystem.

## Introduction

Arctic and sub-arctic marine ecosystems are changing much faster than predicted [1]. Since the advent of satellite records in 1979, sea-ice areal extent has diminished by about 50% at the September minimum, with a roughly 75% year-round reduction in thickness of multi-year ice. This fundamental shift has not been linear in the Pacific Arctic region; rather, there were dramatic step-changes of sea-ice loss in late summer 2007 and 2012, and in winter 2017, 2018 and 2019 near Bering Strait [2]. Ocean temperatures have risen across the Arctic, driven both by increased solar insolation that is no longer reflected back into the atmosphere by sea ice and by transport of warm ocean water from the south into sub-arctic and arctic regions [3]. In the Pacific Arctic region, the loss of sea ice has been accompanied by ocean warming and freshening [4, 5], with 2014–2018 marking a period of increased ocean-atmosphere heat exchange coincident with unprecedented low ice cover [6]. The combination of sea-ice loss and warmer seawater has reset the clock on ecological processes in the Pacific Arctic [7], with the Bering Strait region described as in a state of transformation [8]. Compared to the late 1990s, primary production is initiated earlier in spring, with enormous blooms sometimes encountered under thin sea ice resulting in an overall 57% increase in net productivity [9]. Changes in primary productivity vary at regional and local scales, with production in the Chukchi Sea the highest in the Pacific Arctic region [10]. The biophysical impacts of reduced sea ice, increased ocean temperatures and primary production, combined with shifting atmospheric and ocean dynamics, can drive swift and fundamental changes near the base of marine food webs, the trophic level important to gray whales.

Gray whales are unique among mysticete whales in that they can suction sediment from the sea floor and effectively sieve out the infaunal prey on their short, coarse baleen leaving distinctive mud plumes at the surface (Fig 1A). Gray whales are also capable of efficiently feeding on epi-benthic prey swarms, pelagic zooplankton aggregations (Fig 1B), and even fish roe at the sea surface [11]. In the northern Bering and Chukchi seas, gray whales commonly feed on infaunal amphipods [12, 13], although epi-benthic and surface swarms of euphausiids (*Thysanoessa* spp.; hereafter, krill), eurisid amphipods (*Pontogeneia makarovi*), or cumaceans (*Diastylus glabra*) are sometimes the targeted prey [14, 15]. Reports on fine-scale feeding behavior at coastal study sites offshore Vancouver Island, Canada describe the ease with which gray whales can switch between various prey species based upon availability and sometimes size [16, 17]. While apex predators, such as marine mammals and birds, are now commonly recognized as ecosystem sentinels [18], gray whales were one of the first cetacean species so described. Specifically, evidence of connections between changes in gray whale phenology and distribution with shifts in their environment were summarized for six environmental factors, including oceanographic indices (i.e., Pacific Decadal Oscillation and El Nino Southern Oscillation), sea ice loss (N Bering) and thinning (W Beaufort), and shifts in infaunal and epi-benthic prey availability [19]. Taken together, these observations made a compelling case for gray whales as effective sentinels of ecosystem alteration in North Pacific and western Arctic ecosystems.

The Distributed Biological Observatory (DBO) was initiated in 2010 to provide standardized sampling to investigate biological responses to the rapid physical changes ongoing in the

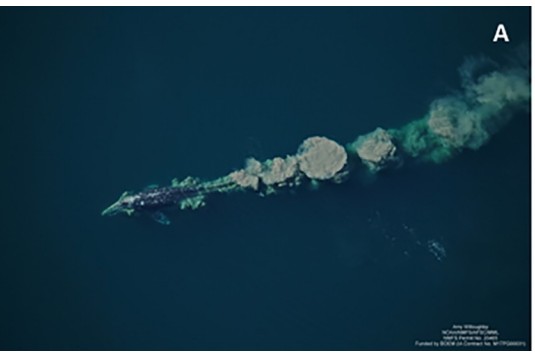
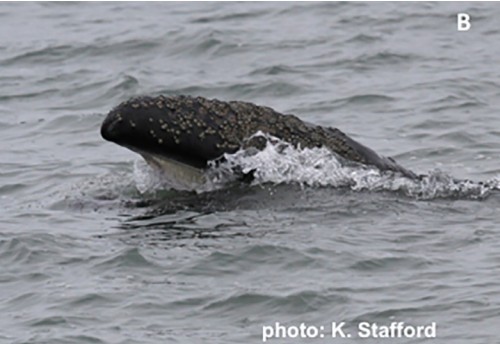

**Fig 1.** Gray whale feeding on infaunal amphipods, resulting in mud plumes (A), and a gray whale skim feeding on krill near Pt. Barrow, Alaska (B). Photo credits: A. Willoughby, NOAA/NMFS/AFSC/MML, NMFS permit number 20465, ASAMM project funded by BOEM via IA M17PG00031 (A); K. Stafford, co-author (B).

Pacific Arctic marine ecosystem (Fig 2A) [20]. An opportunistic marine mammal watch was included in the standard DBO protocol to assess the capacity of marine mammal and other upper trophic level (UTL) species to act as sentinels of ecosystem variability and reorganization [21, 22]. Like fishes and seabirds, marine mammals rely on finding dense aggregations of prey to forage successfully. As a result, shifts in their ecology (i.e., phenology, distribution, and abundance) can signal changes in marine ecosystem trophic structure, which are in turn reflected physiologically by changes in diet and body condition [23]. In addition to sightings during DBO cruises, a robust program of marine mammal research has been conducted in the Pacific Arctic region, comprised of year-round Passive Acoustic Monitoring (PAM) of species-specific calls [24] and seasonal broad-scale Aerial Surveys of Arctic Marine Mammals (ASAMM) [25]. Changes in baleen whale phenology and seasonal distribution have been described based upon some of these data, with correlations to biophysical processes [26, 27] and details on prey availability included when possible [13, 28].

Here, we identify changes in gray whale phenology and seasonal distribution, based upon a compilation of information from the DBO, PAM, and ASAMM programs. We then relate

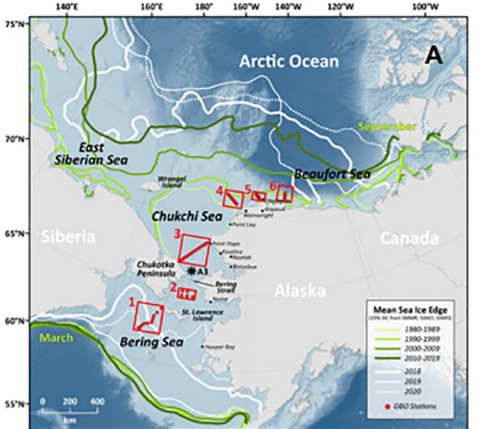
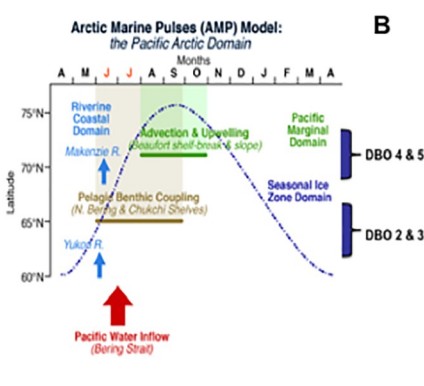

**Fig 2.** The Distributed Biological Observatory (DBO) and mean sea ice edge in the Pacific Arctic (A, revised from [10]), and a schematic of the conceptual Arctic Marine Pulses model (B, revised from [29]) depicting links among biophysical aspects of the Pacific Arctic marine ecosystem. The mean sea ice edge depicts the 15% concentration threshold using SMMR, SSM/I and SSMIS satellite data. All sea ice edge contours north (south) of Bering Strait represent September (March) conditions for three decadal periods (green) and annually for 2018, 2019 and 2020 (white).

those shifts to changes in ocean biophysics that likely impact availability of their prey, including pelagic-benthic coupling, advection, ocean warming and freshening, and large and local-scale wind forcing. The conceptual Arctic Marine Pulses (AMP) model combines these biophysical factors into a regional framework (Fig 2B) [29]. We use the AMP model to contrast the advection and pelagic-benthic coupling drivers active in the Bering Strait region (DBO regions 2 & 3) to those factors <u>combined</u> with seasonal sea ice retention and wind-forcing dynamics in the northeastern Chukchi Sea (DBO regions 4 & 5) to investigate how these processes influence gray whale prey availability. Notably, the inclusion of long-term measures of infaunal crustacean abundance and species composition in DBO regions 2–5 provides direct evidence of, and fosters insights into, the impacts of shifting ocean biophysics on gray whale prey. The potential effects of observed and inferred prey alteration on gray whale nutrition are discussed in the context of an unusual mortality event that began in 2019 resulting in *ca*. 10-fold increase in annual number of stranded gray whales [30]. We close with suggested hypotheses that might guide future research on the ecology of gray and other baleen whales and enhance their capacity to act as sentinels of ongoing transformation of the Pacific Arctic marine ecosystem.

## Methods

### Gray whale passive acoustic and visual sampling

Our assessment of gray whale phenology, defined here as timing of arrival and departure from the Bering Strait region, is based on detection of their distinctive knock-like 'bongos' or 'Class 1 calls' [31]. Gray whale 'moans' or 'Class 3 calls' [31] were also included when detected with 'bongo' calls and when there were no accompanying humpback sounds with which they could be confused. Data were recorded on instruments deployed in or near DBO regions 2 and 3 (NM1 and PH1 on Fig 4A) from 2012–2019 as part of an extensive marine mammal PAM program [24]. Instruments were set to record on a duty cycle of 30% to extend battery life for a full year, with a sample rate of 16384 Hz for an effective bandwidth of 10–8192 Hz, which is sufficient to record all known gray whale signals [31]. Daily call detections were binned in 10-minute increments and normalized by recording effort, resulting in call histograms which depict the percent of calling activity/day, rather than an actual count of calls recorded. We examined histograms of gray whale calling activity at each site to identify phenological changes in annual pattern.

Our summary of gray whale distribution is based on sightings made during: (a) marine mammal watches on 22 DBO cruises (S1 Table), and (b) the broad scale ASAMM program conducted in the northeastern Chukchi Sea from 2009–2019 [25]. On DBO cruises, a visual watch for marine mammals was conducted during daylight hours when the ship was transiting between sampling or mooring stations, augmented by scans around the ship each hour when the ship was on station. An observer trained in marine mammal species identification used naked eye and handheld binoculars to scan a 120˚ arc forward of the ship (abeam, to +30˚ of the bow) out to the horizon. When two people were available to stand watch, the full 180˚ arc forward of the ship was scanned to the horizon. The watch stander was often assisted by other scientific party personnel and the ship's crew. The watch was curtailed when sea state exceeded Beaufort 05 (wind speed ~25kts, 12.8 m/s), or visibility was reduced to < 1km by precipitation or fog. Although DBO cruise tracks and ship speeds were similar among years, sighting rate (whales/km) could not be calculated due to variability in watch effort. Thus, while sightings from DBO cruises provide data on gray whale presence, especially in the Bering Strait region, these data are not included in statistical analyses.

During the ASAMM program, line-transect aerial surveys were conducted from July through September using twin engine aircraft outfitted with left- and right-side bubble windows. The study area encompassed the northeastern Chukchi Sea from 67˚N to 72˚N, and east of 169˚W, an area inclusive of DBO 3–5. Dedicated observers stationed at each window reported all marine mammal sightings and associated environmental conditions to a data recorder. Sightings of large whales were briefly circled over to confirm species identity, group size, behavior, and presence of calves. Surveys were conducted whenever weather conditions allowed (i.e., sea state ≤Beaufort 05; cloud ceiling consistently >335 m). Offshore transect lines were oriented perpendicular to the coastline to allow sampling across isobaths and prevailing currents, and to best assess marine mammal density gradients; a coastal transect was flown *ca*. 1-km offshore to better document nearshore habitat.

To examine changes in gray whale distribution in the context of environmental variability, ASAMM data were divided into three geographic areas for analysis: (1) the southern Chukchi Sea, referred to as the Hope Basin area and inclusive of DBO 3; and the northeastern Chukchi Sea, referred to as the (2) Wainwright and (3) Peard Bay areas, inclusive of DBO regions 4 and 5. Gray whale sighting rates were calculated for each area as the number of gray whales seen/ offshore transect kilometer by month (July, August, September) and year (S2 Table). Effort and sightings on the coastal transect were not included in monthly SR analyses to avoid bias towards nearshore areas. All ASAMM data are publicly available at fisheries.noaa.gov/ resource/data/1979-2019-aerial-surveys-arctic-marine-mammals-historical-database; databases used for analyses include versions 1979_2011_v3_36, 2012_2014_v0_28, 2015_2017_v22, and 2018_2019_v6.

## Gray whale prey and environmental variability

Environmental factors relevant to the influence of pelagic-benthic coupling on gray whale infaunal prey were evaluated based upon a multi-decadal record of macrobenthos and sediment dynamics [32], with special focus on stations dominated by crustacean species at the start of the 2010–2019 time series [32, 33]. In brief, stations in DBO regions 2–5 were designed to sample sediments across localized areas of high benthic productivity and biomass. Stations in DBO regions 3, 4 and 5 were oriented along diagonal transects, while those in DBO 2 sampled along latitudinal transects. South of DBO 3, a series of stations (designated UTN, for University of Tennessee), were oriented roughly parallel to the International Date Line (IDL) from 66.5°N to 68°N latitude. Over the 2010–2019 period, small adjustments were made to sampling design in DBO regions 2 and 4. Specifically, In the DBO 2 region, the four original times series stations (UTBS) were augmented from 2016 onwards by the addition of three stations (BCL6c, DBO2.7, UTBS2A) to expand the spatial extent of sampling. In the DBO4 region, station placement was changed from six stations along a single transect line to six stations on three shorter transect lines (shown as DBO4 O, N, and n; Fig 5) to improve sampling the patchy distribution of benthic fauna in that area. At all stations, replicate sediment samples were collected using a 0.1 m$^2$ weighted van Veen grab. Sediments were sieved onboard through 1 mm mesh screens, with the retained macrofauna preserved with 10% buffered seawater and formalin for postcruise taxonomic identification and analysis of abundance and wet weight biomass. The analysis of infaunal prey for gray whales focused only on the Class Crustacea, which is comprised primarily of amphipods, with a very minor (< 1%) inclusion of isopods and cumaceans. Additional details of sampling and analysis methods are provided in [32].

Gray whale infaunal crustacean prey were analyzed with attention to changes in species composition and abundance in response to trends in sea ice loss, ocean temperature, salinity, and measures of chlorophyll *a* both in the water column and in surface sediment. Spatial

interpolation was accomplished using geographical information system software [34]. Specifically, the Geostatistical Analyst Wizard Inverse Distance Weighting (IDW) tool in ESRI's Arc-GIS Desktop v.10.8.1 was used with default settings to produce an interpolated surface map based on abundance of the class Crustacea per station for the years 2010 to 2019. Temporal patterns in crustacean abundance and Spearman's rho rank correlation analysis was used to determine correlations of crustacean abundance over time using JMP$^{TM}$ Pro 15.2.0 (www.jmp.com). Most macrofaunal data sets for the 2010–2019 period are available at the Arctic Data Center (ADC) DBO project page (https://arcticdata.io/catalog/portals/DBO/Data)), and the NOAA National Centers for Environmental Information (NCEI) website (https://www.ncei.noaa.gov/), although some replicate samples could not be included due to Covid limitations on laboratory work. To clarify data included this paper, a summary of crustacean abundance is listed by cruise number, station and date is provided supplemental S3 Table. Of note, the environmental data used for statistical analyses in relation to crustacean abundance listed in S3 Table are available at the aforementioned ADC and NCEI data archives for the associated cruises.

Environmental factors potentially indicative of the influence of advection and circulation on gray whale epi-benthic and pelagic prey availability were evaluated through application of an iterative correlation analysis (ICA) method [35, 36]. The ICA method is a supervised machine learning tool that uses, in the present case, a multiyear time series of gray whale SR derived from ASAMM data as a training set to identify a statistically similar interannual time series of either (i) a seasonally-averaged environmental factor (e.g. volume, heat, freshwater transports) measured at the 'climate mooring' north of Bering Strait [4; mooring A3], or (ii) local- to large-scale (Bering-Chukchi-Beaufort region) wind regimes derived from daily sea level wind products available at the National Centers for Environmental Prediction (NCEP)/National Center for Atmospheric Research [37]. For an environmental factor with time series information at a single location (e.g. transport at mooring A3), ICA identifies the seasonal start and end dates for which correlations between seasonally-averaged transports and gray whale SRs are statistically significant. For an environmental factor with time series information at multiple locations (e.g. NCEP winds), ICA identifies the seasonal start and end dates that maximize the ocean area north of the Bering Sea shelf break over which correlations between seasonally-averaged winds and gray whale SRs are statistically significant. Results for both single and multiple location ICA are summarized in the form of a heat map matrix that identifies the best-fit start and end dates from a range of candidate seasonal averaging periods.

## Results

### Changes in gray whale phenology and distribution

**Acoustic detections.**   A change in gray whale acoustic activity was observed in the northern Bering Sea (DBO 2) where the period of calling activity extended from late May through November in 2012–2015, shortened to late May to mid-September in 2016, and followed by a near cessation of call detections in 2017–2019 (Fig 3A). In the southern Chukchi Sea (DBO 3), consistent calling activity extended from mid-June through October in 2012–2015. In 2017 call detections ended a month earlier (September), and there was a near cessation of calling activity throughout 2018–2019 (Fig 3B). These changes in calling activity suggest that gray whales departed DBO regions 2 and 3 earlier each year after 2016 and 2017, respectively.

**Visual detections.**   Gray whales were seen during marine mammal watches on DBO cruises in all years (Fig 4A). Areas where gray whales were commonly seen included the northern Bering and southern Chukchi Seas, and waters offshore Wainwright and along the coast between Pt. Franklin and Pt. Barrow. While DBO cruise tracks were similar among years,

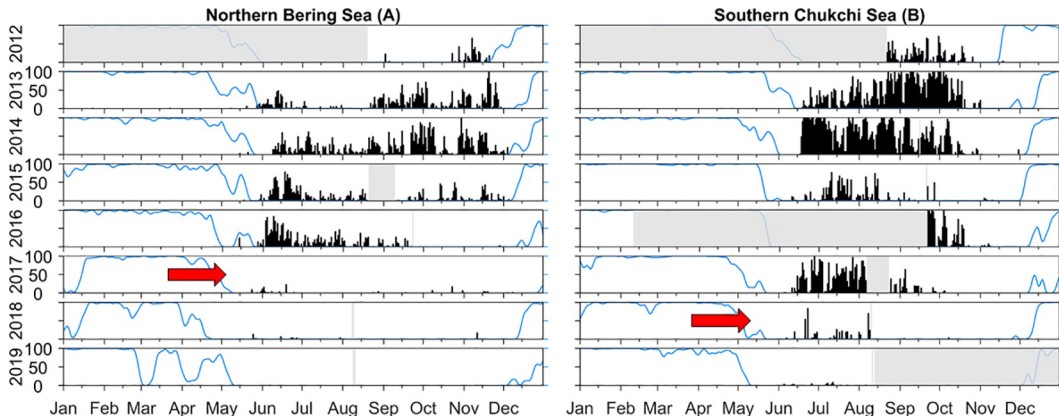

**Fig 3.** Gray whale annual calling activity in the northern Bering (A) and southern Chukchi (B) seas, 2012–2019. Black bars show gray whale calling activity, blue lines represent seasonal ice cover and grey shading indicate periods for which data are unavailable. Red arrows denote dramatic drop-off in calls coincident with 2017–2019 winter sea ice loss event. Hydrophone locations are shown in Fig 4 - DBO map.

timing and platform operations varied, limiting inferences that can be drawn from these data for lack of SR calculations. Of note, however, is that gray whales were consistently seen in the northern area of DBO 2 and near the central and western-most sampling stations in DBO 3 (Fig 2A), as well as west of the International Date Line (IDL) in years that ships had access to those waters (2009, 2010, 2012). Also, in contrast to the PAM data, gray whales were seen in DBO regions 2 and 3 in 2017–2019 after the aforementioned loss of winter sea ice in those areas.

The ASAMM data provide the means to relate gray whale distribution to survey effort north of 67°N and east of 169°W in the Chukchi Sea. Over the eleven-year study period, there were 1,333 sightings of 2,358 gray whales from July through September, with most whales seen in south-central and northeastern Chukchi Sea waters (Fig 4B). Interannual variability in SR

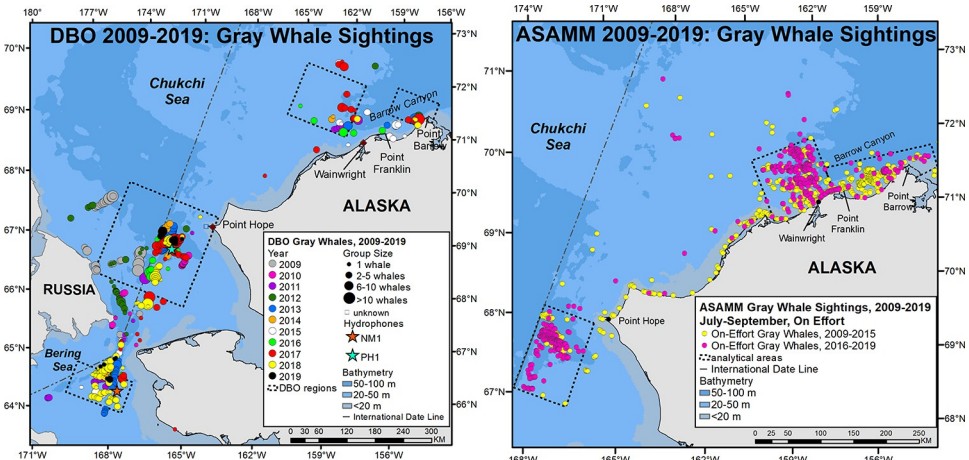

**Fig 4. Gray whale distribution from sightings made during marine mammal watches on DBO cruises, and the broad scale Aerial Surveys of Arctic Marine Mammals (ASAMM) program, 2009–2019.** Dashed lines depict boundaries of DBO regions and analytical areas, respectively. Gray whale sighting rates (SR = whales/km) in analytical areas were calculated solely from ASAMM data. ASAMM data are publicly available at fisheries.noaa.gov/resource/data/1979-2019-aerial-surveys-arctic-marine-mammals-historical-database. Databases used for maps include versions 1979_2011_v3_36, 2012_2014_v0_28, 2015_2017_v22, and 2018_2019_v6.

**Table 1. Annual ASAMM survey effort (km), number of gray whale sightings, number of whales seen and sighting rate (SR = whale/km), July-September, 2009–2019.** Includes all sightings and effort on transect, including coastal and offshore transects, in ASAMM study area.

| Year | Effort km | No. sightings | No. gray whales | SR (whales/ km) |
|------|-----------|---------------|-----------------|-----------------|
| 2009 | 15,560 | 88 | 115 | 0.007 |
| 2010 | 16,089 | 52 | 74 | 0.005 |
| 2011 | 20,010 | 102 | 139 | 0.007 |
| 2012 | 21,077 | 124 | 221 | 0.011 |
| 2013 | 21,768 | 67 | 110 | 0.005 |
| 2014 | 19,032 | 164 | 291 | 0.015 |
| 2015 | 20,865 | 98 | 185 | 0.009 |
| 2016 | 21,381 | 161 | 366 | 0.017 |
| 2017 | 23,021 | 207 | 384 | 0.017 |
| 2018 | 15,589 | 167 | 311 | 0.020 |
| 2019 | 13,121 | 103 | 162 | 0.012 |
| Total | 207,513 | 1,333 | 2,358 | 0.011 |

for the entire study area ranged from a high of 2 whales/100km in 2018 to a low of 0.5 whales/100km in 2010 (Table 1). In 2015, there was a *ca*. 110 km (60 nm) shift in gray whale distribution away from the coastal habitat between Pt. Franklin and Pt. Barrow to waters offshore Wainwright and near the head of Barrow Canyon. In the south-central Chukchi Sea, gray whale distribution clustered in a benthic trough 'hotspot' southwest of Pt. Hope in all years, with sightings extended further south after 2015. Since gray whales spend much of their time feeding while in the Chukchi Sea, these changes in distribution and associated shifts in SR were examined (i) with reference to observed changes in infaunal crustacean prey abundance and community composition from benthic sampling, and (ii) by correlation of sighting rates from ASAMM data with environmental factors associated with availability of epi-benthic and pelagic prey, as described in the next two sections.

## Changes to infaunal crustacean abundance, community composition, and associated environmental factors

Our multi-decadal record of infaunal crustacean abundance provided the means to directly track changes to gray whale infaunal crustacean prey in productivity hotspots in the northern Bering and eastern Chukchi seas (Fig 5). The area of highest gray whale prey abundance is in the south-central Chukchi Sea, near the northernmost UTN and the western-most DBO3 stations and extends west of the International Date Line (IDL). Two smaller centers of prey abundance occur in DBO regions 4 and 5 in the northeastern Chukchi Sea. Conversely, the DBO 2 region, which used to encompass a well-documented gray whale prey hotspot in the northern Bering Sea, now is relatively 'cool' with only moderate measures of infaunal crustacean abundance.

The clearest change in gray whale prey during the 2010–2019 period was a significant decrease (r = -0.314, p≤0.05) in amphipod abundance in the DBO2 region (Table 2; Fig 6). Especially notable was the decline in crustacean abundance at the northwest sampling sites starting in 2015 and the nearly complete absence of crustacean prey at southwest sampling stations after 2012 (Fig 7). Of note, this decline in crustacean abundance represents a continuation for gray whale prey loss this region, which began in the late 1990s [32].

In the southern Chukchi Sea, comparatively small-bodied amphipod species from three families were found in the DBO 3 region, where gray whales are often seen feeding from both vessel and aircraft platforms. There was no significant trend in infaunal crustacean abundance

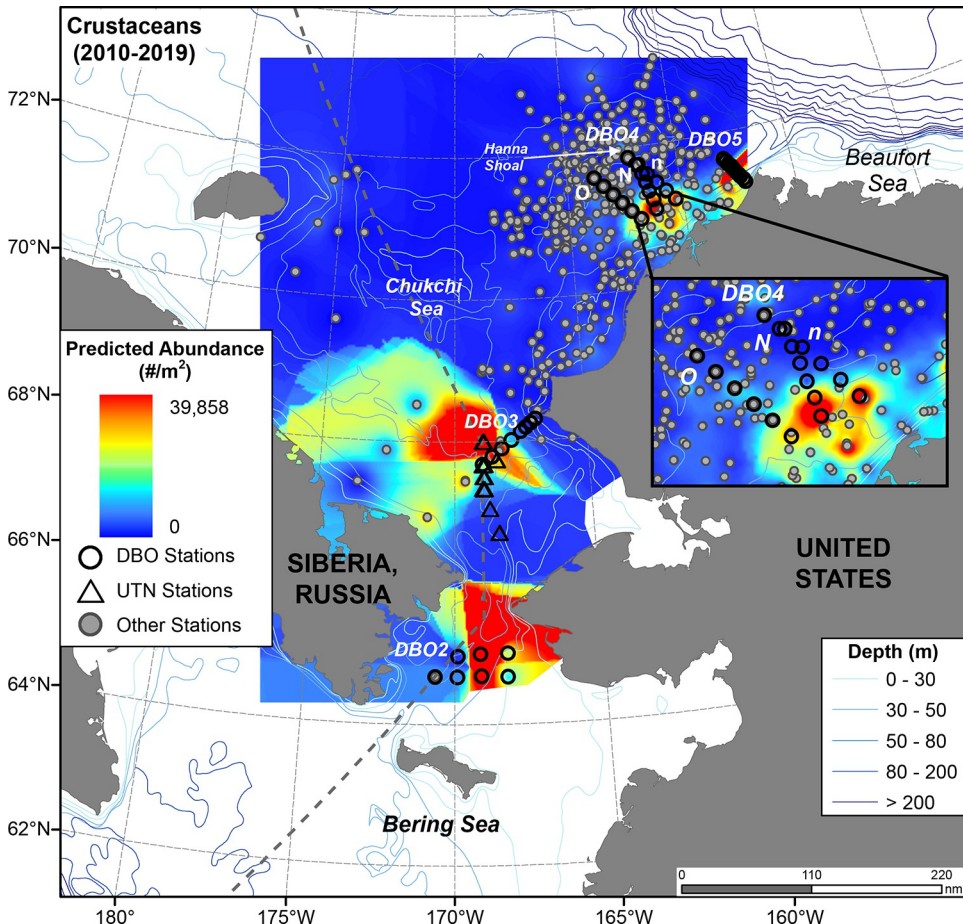

**Fig 5. Distribution of infaunal crustacean abundance (#/m²) at all stations where macrofauna were sampled during the period 2010–2019 in the northern Bering and eastern Chukchi seas.** Key: open circles = DBO stations, open triangles = UTN stations, and closed gray circles = all remaining stations. Note that DBO4 has 3 transect links: O = original line 1, n = new line 1, and N = new line 2. The spatial interpolation map was made using the Geostatistical Analyst Wizard Inverse Distance Weighting (IDW) tool in ESRI's ArcGIS Desktop v.10.8.1 (ESRI 2020) with default settings. See S3 Table for DBO sampling site information and crustacean abundance data. The remainder of the all station crustacean abundance data are available at Arctic Data Center DBO data portal <https://arcticdata.io/catalog/portals/DBO/Data>.

in this region, with the highest crustacean abundance consistently observed in the hotspot sampled by the UTN 7 and SEC 3 and 4 stations (Figs 6 and 7). Notably, although abundance remained stable over time, there was a shift in community composition, whereby the northernmost stations of the DBO 3 hotspot are now dominated by *Pontopoporeia femorata* (F. Pontoporeidae), which replaced the *Byblis* sp. (F. Ampelicidae) in the mid-2000s and has remained the dominant crustacean species since that time; meanwhile the small F. Isaeidae amphipods have remained second in abundance throughout the last decade.

In the northeastern Chukchi Sea, feeding gray whales were seen where both comparatively large infaunal amphipods (F. Pontoporeidae) and a variety of small infaunal crustacean species (F. Isaidae., F. Phoxochelidae) were found in a localized area along the eastern flank of Hanna Shoal in DBO 4 (Fig 5, Table 2). The infauna at all 6 stations in the time series were equitable in abundance (Fig 6), but there was notably higher amphipod abundance in 2016 and 2017 at the DBO 4.2N station (Fig 7) on the east flank of Hanna Shoal and close to the head of Barrow Canyon. In DBO 5, both the larger amphipods (F. Ampeliscidae) and smaller amphipods (F.

**Table 2. Changes in infaunal crustacean community composition and abundance coincident with bottom water temperature and salinity, sea ice persistence (days from Sept-to Sept), and integrated water column and sediment chlorophyll in DBO regions 2–5 for the period 2010–2019.**

| Parameters | REGION | | | |
|---|---|---|---|---|
| | DBO2 | DBO3 | DBO4 | DBO5 |
| | 2010–2019 | 2010–2019 | 2010–2019 | 2010–2019 |
| Dominant crustacean families (by abundance) | Ampeliscidae, Isaeidae | Ampeliscidae, Isaeidae, Pontoporeidae | Isaeidae, Phoxocephalidae, Pontoporeidae | Ampeliscidae, Isaeidae, Phoxocephalidae, Leuconidae |
| Crustacean Abundance (no/m2) | 5546 ± 6610 | 3588± 5477 | 3454 ± 4999 | 5400 ± 7528 |
| | **r = -0.314*** (51) | r = -0.046, ns (134) | r = 0.065, ns, (51) | r = -0.022, ns (61) |
| Bottom water temperature (˚C) | 1.98 ± 1.20 | 3.34 1.82 | -0.57 ± 1.43 | -0.41 ± 1.89 |
| | **r = +0.678**** (51) | **r = +0.377**** (134) | **r = +0.616**** (51) | r = -0.072, ns (61) |
| Bottom water salinity (psu) | 32.56 ± 0.31 | 32.28 ± 0.51 | 32.44 ± 0.29 | 32.50 ± 0.85 |
| | **r = +0.367**** (51) | r = +0.113, ns (134) | **r = -0.502**** (51) | r = +0.184, ns (61) |
| Sea ice persistence (days/year) | 143 ± 28 | 172 ± 18 | 221 ± 32 | 260± 15 |
| | **r = -0.756**** (51) | **r = -0.725**** (134) | **r = -0.896**** (51) | **r = -0.634**** (61) |
| Integrated water chlorophyll *a* (mg/m2) | 66.73 ± 53.43 | 162 ± 183 | 106.80 ± 106.23 | 115.63 ± 129.64 |
| | **r = +0.370**** (51) | r = +0.131 (134) | **r = -0.296*** (51) | r = -0.056, ns (60) |
| Sediment chlorophyll *a* (mg/m2) | 19.54 ± 7.24 | 20.55 ± 9.02 | 14.49 ± 7.80 | 13.65 ± 7.63 |
| | **r = +0.399**** (51) | r = -0.058, ns 134) | **r = +0.533**** (51) | r = -0.117, ns, (56) |

Values are average ± s.d. Key: **bold** is significant increase (+) or decrease (-) over the decade

*p<0.05

**p<0.01; ns = not significant; (#) = number of stations, with sea ice persistence in number of days/year. Dominant species/family: Ampeliscidae (*Ampelisca macrocephala*, *A. birulai*, *Byblis* sp.), Isaeidae (*Protomedeia fasciata*, *Photis* sp.), Leuconidae (*Eudorella pacifica*), Phoxocephalidae (*Grandifoxus* sp., *Heterophoxus* sp.), Pontoporeidae (*Pontoporeia femorata*).

Isaeidae and F. Phoxocephalidae) and cumaceans (F. Leuconidae*)* were abundant along the narrow shelf nearshore (Table 2). Although there was no significant trend in infaunal crustacean abundance in the DBO 5 region (Fig 6), there was an indication of a local increase in abundance near the head of Barrow Canyon at the BarC 6 station in 2015 and especially in 2017 (Fig 7). Unfortunately, sea ice cover in 2018 precluded sampling the DBO5 stations and sorting of 2019 samples is in progress.

A summary of changes in environmental factors associated with the sampling of infaunal crustaceans provides context, but no simple answers, as to why there was a significant change in abundance only in the DBO 2 region (Table 2). Sea ice persistence has *decreased* in all DBO regions 2–5, while integrated water column chlorophyll increased only in the DBO2 region, suggesting an enhanced pelagic food supply in waters north of St. Lawrence Island. In contrast, seasonal variability in water mass nutrient content in the eastern Chukchi Sea (i.e., DBO 3–5) influences chlorophyll and productivity measurements over the open water season (see Frey et al, this volume). Notably, sediment chlorophyll increased only in DBO 2 and DBO 4. This suggests food supply to infaunal crustaceans has increased in those regions, making the *decline* of amphipods in DBO 2 unexpected and indicating that some combination of environmental factors is driving abundance. In reviewing those factors, we note that bottom water temperature has increased in all regions except DBO 5, accompanied by increased bottom water salinity in DBO 2 (Table 2). Bottom water salinity was variable across the DBO3 transect from nearshore to offshore stations, resulting in no trend. Bottom water became significantly fresher in DBO 4, likely due to increasing sea ice melt and perhaps freshwater pulses from the south. There was no change in bottom water salinity in DBO 5, likely due to the seasonal complexity of Barrow Canyon ocean dynamics.

 

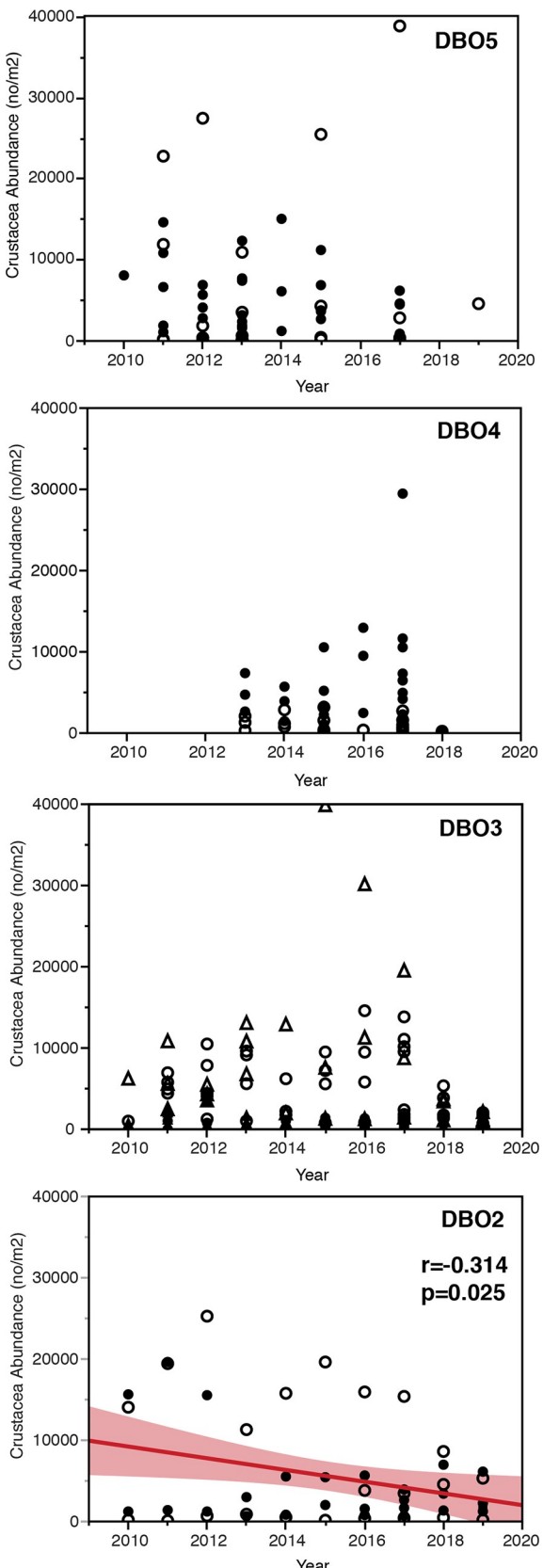

**Fig 6. Macrofaunal crustacean abundance at time series stations over the 2010–2019 period in DBO regions 2–5 regions.** A: DBO2 in the Northern Bering Sea (squares), with the red line indicating the linear fit using all the station data, with the confidence curves shaded red around the line and the correlation coefficient indicating the significant decline of crustacean abundance. B: DBO3 in the SE Chukchi Sea (triangles), C: DBO4 in the NE Chukchi Sea (circles), and D: DBO5 in Barrow Canyon (diamonds) had no significant trends in crustacean abundance over time, although there was spatial variation in values. Key: within each DBO region, the closed symbols = southern time series stations; open symbols = northern time series stations. This format provides a spatial perspective of station location for each DBO region, with reference to Fig 5.

## Environmental factors related to gray whale epi-benthic and pelagic prey availability

Since gray whales also feed on epi-benthic (e.g. mysids, cumaceans [11]) and pelagic (e.g. krill [14]) prey, integrative correlational analyses (ICA) were used to investigate drivers of Chukchi Sea circulation (regional wind forcing and transport through Bering Strait) that can alter prey availability and thereby influence whale distribution and sighting rates (SR). In the northeastern Chukchi Sea, interannual changes in gray whale monthly distribution and SR depict a shift in foraging away from the nearshore Peard Bay area (2009–2014) to offshore waters near the head of Barrow Canyon in the Wainwright area (2015–2019; Fig 8). The shift in distribution and SR is most evident for the month of July, with a near-abandonment of the Peard Bay area in August and September. To investigate this shift, combined annual July-August-September (JAS) SRs for Wainwright (W) and Peard Bay (PB; Fig 9A) were used to create a 2009–2019 time series of fractional SR, ($W_F = W_{JAS}/(W_{JAS} + PB_{JAS})$. This served as the ICA training set (Fig 9B) to identify periods of seasonally-averaged winds within the Bering-Chukchi-Beaufort domain that were significantly correlated ($r > 0.60$, $p<0.05$, df = 9) with the fractional SR. The ICA heatmap shows three loci, all with start dates in early June (Fig 9C), with averaging periods of: (i) ~21 days (end date late June; best fit indicated by diamond symbol), (ii) ~45 days (end date late July), and (iii) ~100 days (end date mid-September). The resulting wind regimes were similar for all three loci in that the action center is the region north of ~70° N and that statistically-significant variability primarily occurs with the E-W component of the wind field. These results suggest foraging conditions for epi-benthic and pelagic prey are relatively better in the Peard Bay area when late-spring and summer winds are easterly and persistent (warm color shading) over the northern Chukchi and southern Beaufort (Fig 9D). Conversely, gray whale foraging on epi-benthic or pelagic prey (e.g. krill) is likely better in the Wainwright area when late-spring and summer winds are weak, variable (cool color shading), or westerly (Fig 9E).

Interannual variability in gray whale monthly distribution and SR in the Hope Basin area depicts a more stable situation. Sightings were clustered around a quasi-stationary front associated with a bathymetric trough southwest of Pt. Hope, with a southern extension of sightings evident in the combined July distribution (Fig 10). Of note, there were very few gray whales seen in 2009–2013, likely due in part to limited survey effort in those years (S2 Table). Except for 2015, the combined annual JAS SR for gray whales in the Hope Basin area was consistently higher from 2014–2019 than for the years 2009–2013. Notably, an ICA for the Hope Basin area showed no clear relationship with winds over the Bering-Chukchi shelf region. Combined monthly SR showed intra and interannual variability, with peaks in July (2017 and 2019), August (2014 and 2017), and September (2014 and 2016) (Fig 11).

Interannual variations in gray whale SR in the Wainwright, Perd Bay and Hope Basin analytical areas were used as ICA training sets to identify time series of monthly-averaged transport at the A3 'climate' mooring site (see Fig 2A) north of Bering Strait. ICA results indicate that winter-to-early-spring volume, heat, and freshwater transports are positively correlated

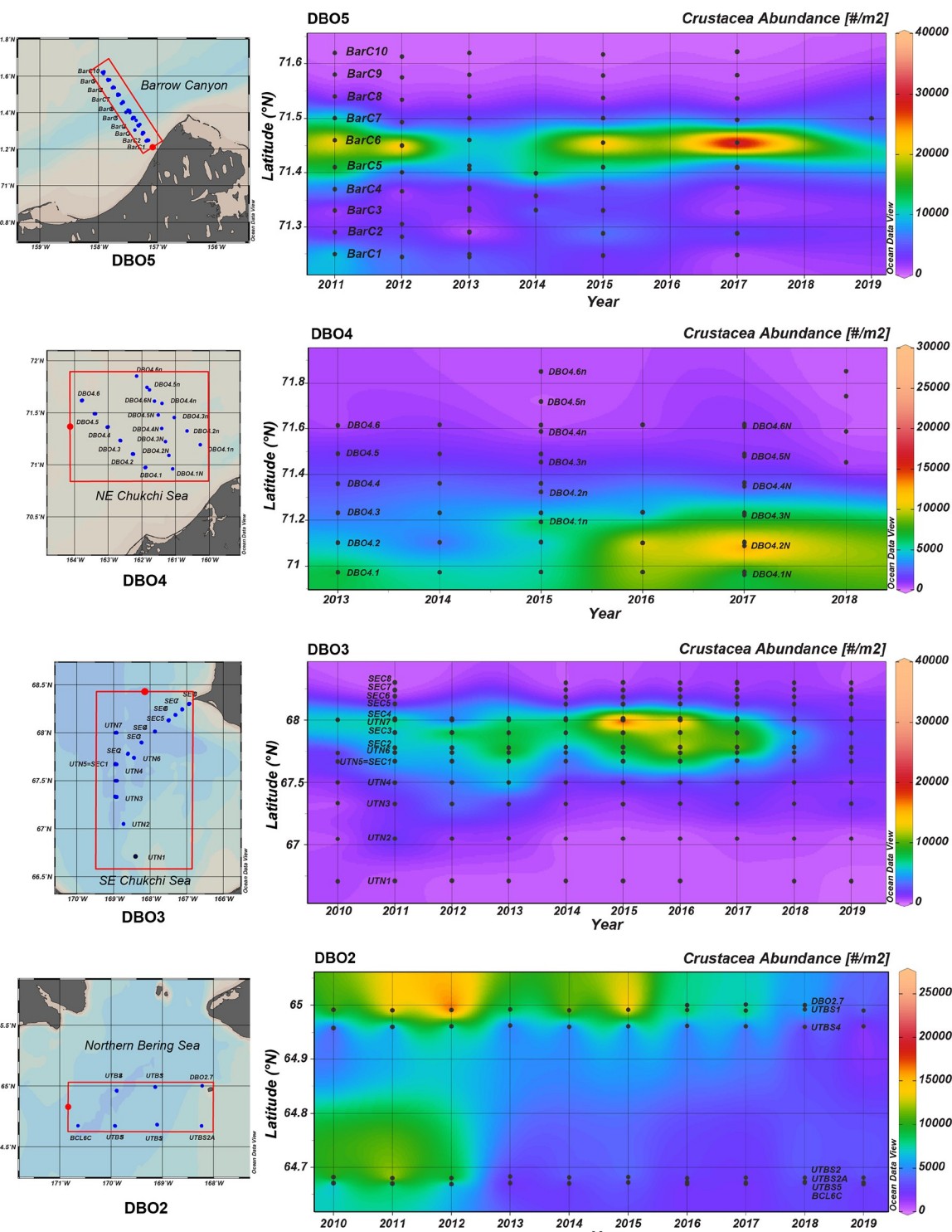

**Fig 7. Time series of macrofaunal crustacean abundance at stations in DBO regions 2–5 over the 2010–2019 period.** Figures are aligned from south to north over a latitudinal gradient with the red bounding boxes surrounding time series stations in each DBO region. The crustacea summary data plotted in this figure are provided in S3 Table and the Arctic Data Center DBO project page (https://arcticdata.io/catalog/portals/DBO/Data). All crustacean data in each DBO region were plotted using Ocean Data View software under a CC By license; see Methods for details of stations included in the visualization.

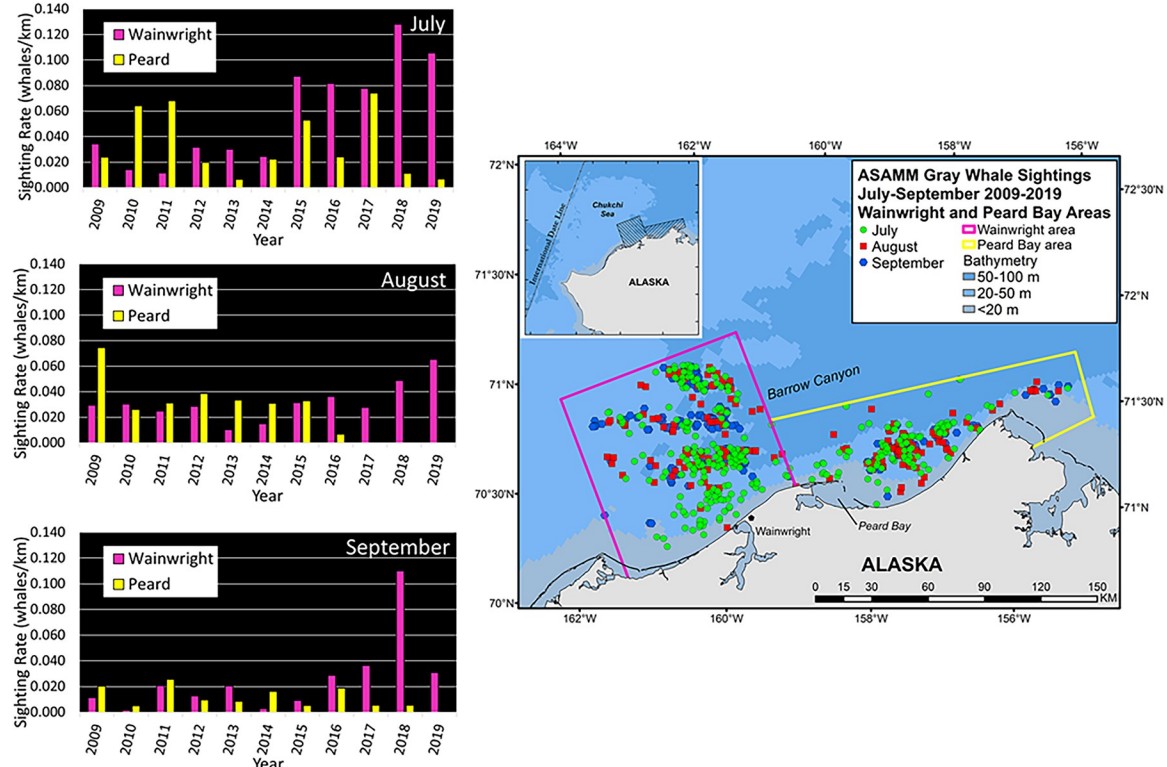

**Fig 8. Interannual variability in gray whale monthly distribution and histogram of sighting rates (SR) for Wainwright and Peard Bay analytical areas in the northeastern Chukchi Sea, 2009–2019.** Inset map shows the location of Wainwright and Peard Bay areas relative to the ASAMM study area. ASAMM data are publicly available at fisheries.noaa.gov/resource/data/1979-2019-aerial-surveys-arctic-marine-mammals-historical-database. Databases used for maps include versions 1979_2011_v3_36, 2012_2014_v0_28, 2015_2017_v22, and 2018_2019_v6.

with the combined JAS SR in the Wainwright area and negatively correlated with combined JAS SR in the Peard Bay area (Table 3). In the Hope Basin area, interannual variations in August and September SR were negatively correlated with volume, heat, and freshwater transports. Notably, the negative correlations for the September SR were associated with 10-12-month time-averages of transport compared to the 1 to 4-month averages of transport found in other comparisons.

A summary of results relating variability in gray whale sighting rates (SR) to circulation drivers (A3 transport and winds), bathymetry, and likely whale prey highlights contrasts among the three analytical areas in the Chukchi Sea (Table 4). Notably, transport at A3 is correlated with gray whale SR in each of the regions, while a contrasting pattern of high and low SR can be correlated with alternate wind patterns only in the Peard Bay and Wainwright areas. In sum, bathymetry and transport appear to be drivers of gray whale pelagic prey in the Hope Basin area (inclusive of the DBO 3 region), while bathymetry, transport and wind forcing play a role in the northeastern Chukchi Sea (inclusive of DBO 4 and 5).

## Discussion

### Historical overview of gray whale ecology

Gray whale ecology in the northern Bering and Chukchi Seas has been the focus of intermittent research since the late 1960s, which provides a foundation upon which to interpret the

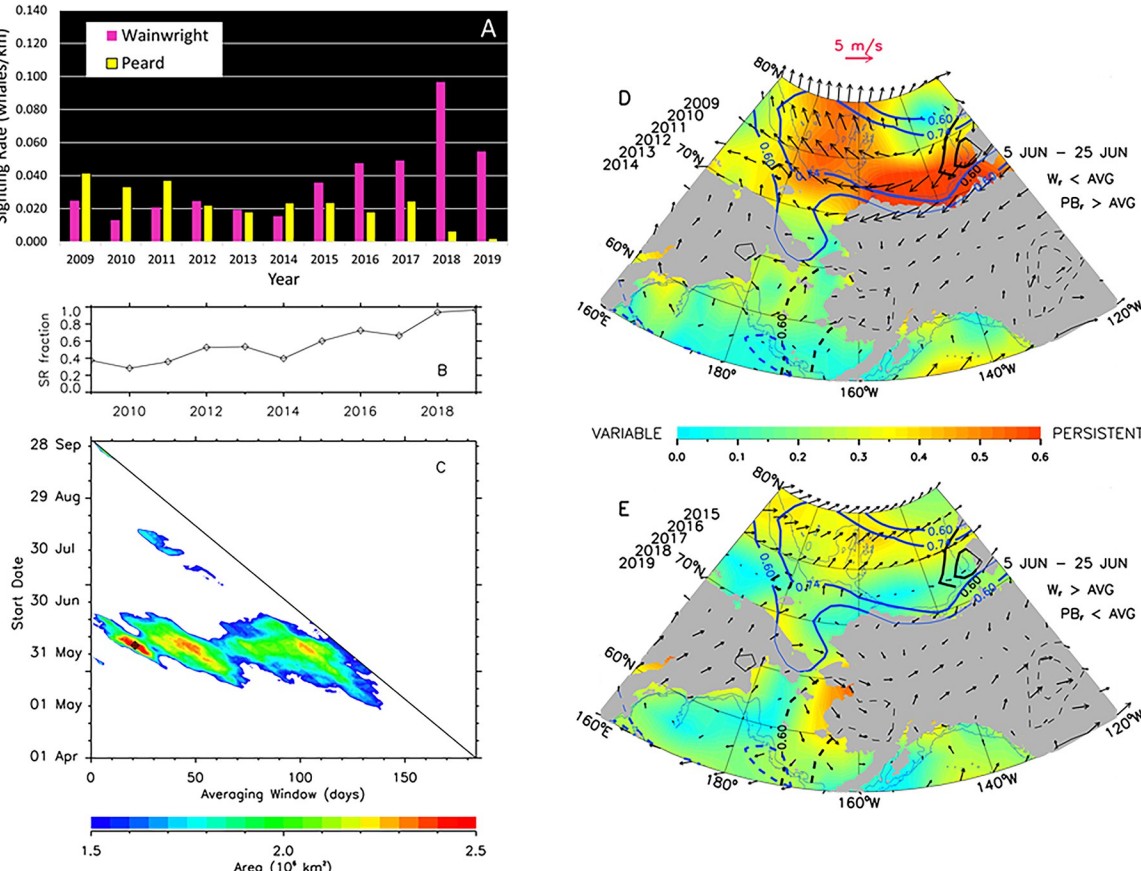

**Fig 9.** Relationship between gray whale sighting rates (SR) and wind forcing, including: Histogram of combined average July-August-September SR in the Wainwright and Peard Bay analytical areas (A); Wainwright area SR fraction (B); ICA heat map showing the area of the Bering-Chukchi-Beaufort ocean domain over which correlations between the SR fraction and time-averaged winds are statistically-significant (r > 0.602, p<0.05; df = 9, two-tailed test) (C); mean atmospheric circulation from 5 June to 25 June for years in which the Wainwright SR fraction was less than average and the Peard Bay SR fraction was greater than average (D); mean atmospheric circulation from 5 June to 25 June for years in which the Wainwright SR fraction was greater than average and the Peard Bay SR fraction was less than average (E). Mean wind vectors plotted at every 2nd ith grid point. Color shading indicates wind directional constancy, with correlations between SR fraction and U-component winds (blue contours) and V-component winds (black contours) statistically significant at r = 0.60 (p < 0.05) and 0.74 (p < 0.01).

changes in their phenology and distribution reported here. For example, maps from Russian surveys from 1968–1982 show a near-continuous gray whale distribution along the Chukotka peninsula, with clustered aggregations of hundreds of whales in the coastal and central south-western Chukchi Sea [38]. Likewise, in a review of Russian research associated with commercial whaling, gray whales were reported as commonly seen feeding on benthic and epibenthic organisms in shallow waters along the Chukotka peninsula where prey concentrations were greater than 100 g/m$^2$, with distribution extending northwestward to the Eastern Siberian Sea and a very large aggregation (~2000 whales) encountered roughly 160 km offshore between 67˚40-68˚15'N and 169˚40'-172˚W during a 1980/81 expedition [12]. Stomach contents from whales taken in Mechigmensky Bay in 1980/81 contained roughly 122 benthic species [39], with various species of amphipods the most common in occurrence by percentage; e.g. *Pontoporeia femorata* (57.9%), *Ampelisca macrocephala* (53.9%), *Byblus longicornis* (51.3%) and *Lembos arcticus* (47.3%) Similarly, a satellite tracking study of 9 gray whales offshore Chukotka in late summer 2006 found that whales remained in shallow (< 30m) nearshore waters (< 5 km from coast) and were likely feeding on dense aggregations of amphipods [40].

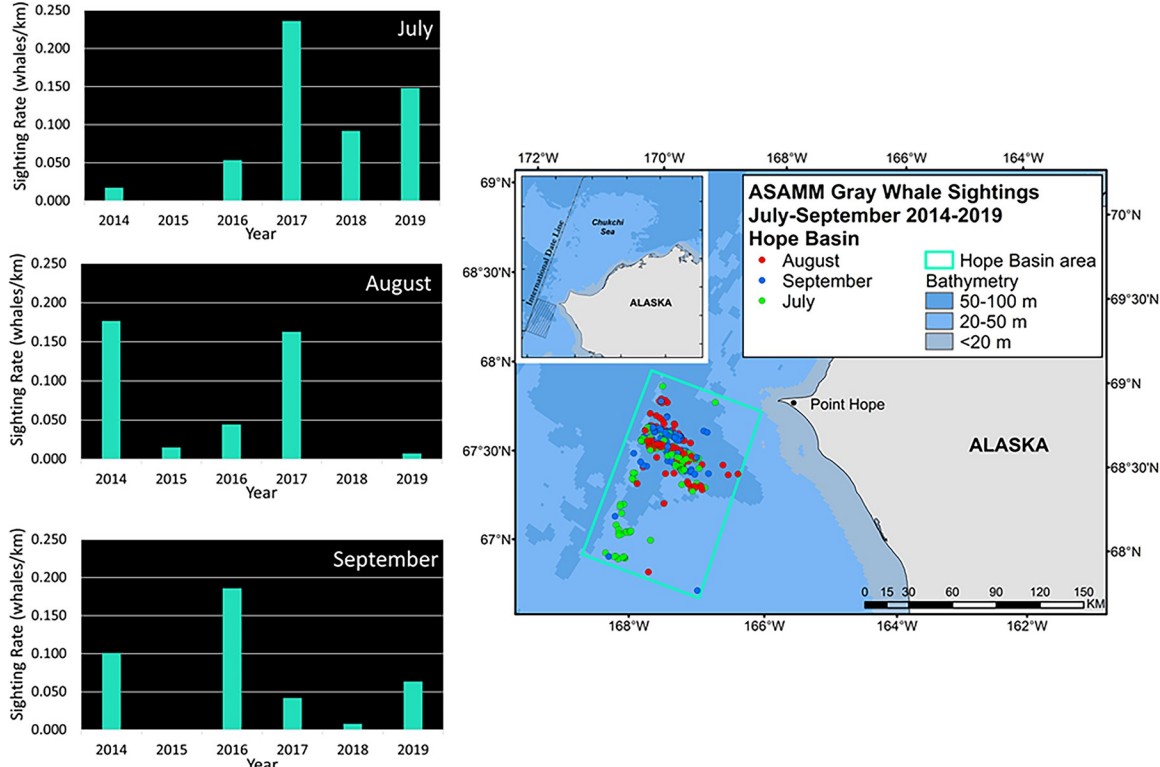

**Fig 10. Interannual variability in gray whale monthly distribution and Sighting Rates (SR) for the Hope Basin area in the southern Chukchi Sea, 2014–2019.** Inset map shows the location of the Hope Basin area relative to the ASAMM study area. ASAMM data are publicly available at fisheries.noaa.gov/resource/data/1979-2019-aerial-surveys-arctic-marine-mammals-historical-database. Databases used for maps include versions 2012_2014_v0_28, 2015_2017_v22, and 2018_2019_v6.

While maps from 1975–1980 surveys offshore Alaska depict comparatively few gray whale sightings in the southeastern Chukchi Sea, there is a distinct distribution cluster in what is now called the DBO 3 'hotspot' area [41]. Conversely, the distribution of gray whales in the northern Bering Sea was extensive, with a dense cluster of sightings extending north from St. Lawrence Island through Bering Strait. This dense summertime distribution of gray whales in the DBO 2 area continued at least through the mid-1980s [42, 43], but by the early 2000s gray whale sightings were comparatively sparse and shifted northward following a shrinking pattern of amphipod distribution [44, 45]. In the northeastern Chukchi Sea, focal areas for gray whale distribution in the 1980s included coastal waters between Icy Cape and Pt. Barrow, with a cluster of sightings near Hanna Shoal northwest of Wainwright [42]. There are no year-round acoustic data for the Bering Strait region in 1980s, but gray whales were reported in the northern Bering and Chukchi Seas from May through at least mid-October, with some whales seen as late as December seemingly following the seasonal cycle of sea ice retreat and formation [12, 41].

## Changes in gray whale phenology

Gray whale phenology, as described by patterns of calling activity in the Bering Strait region, suggests little change in late May arrival times from 2012–2016 in the northern Bering Sea (DBO 2), and through 2017 in the southern Chukchi Sea (DBO 3). Departure dates inferred from a drop in calling activity changed dramatically thereafter, shifting from November to

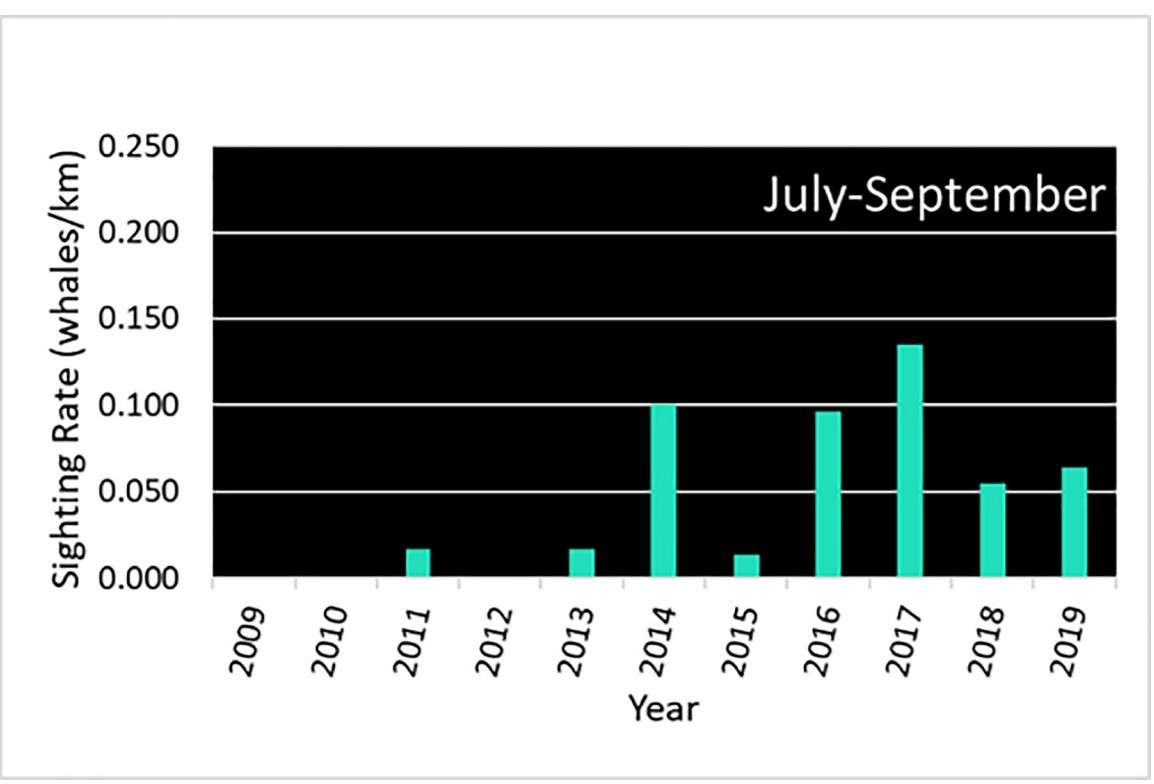

**Fig 11. Gray whale combined July-August-September sighting rates (SR) in the Hope Basin area in the southern Chukchi Sea, 2009–2019.** ASAMM data are publicly available at fisheries.noaa.gov/resource/data/1979-2019-aerial-surveys-arctic-marine-mammals-historical-database. Databases used for maps include versions 1979_2011_v3_36, 2012_2014_v0_28, 2015_2017_v22, and 2018_2019_v6.

mid-September in 2016 (DBO 2) and from October to September in 2017 (DBO 3). This shift towards earlier departure dates and shorter residency periods culminated in a near-absence of calling activity in years 2017–2019 (DBO 2) and in 2018–2019 (DBO 3). This drop in calling activity in each region after 2017 and 2018 suggests that gray whales may have abandoned formerly prime foraging areas coincident with the anomalous 2017–2019 winter sea ice loss event [46]. Alternatively, because gray whale acoustic activity was based on detection of only two call types [31, Class 1 and 3], the drop in call detections could reflect changes in age-class composition or behaviors associated with those signals, rather than whale departure.

**Table 3. Coefficients of linear correlation between 1 to 12- month time-averaged transport at mooring A3 (Woodgate 2018) and gray whale July-August-September (JAS) sighting rates (SR).**

|  | Volume transport | Heat transport | Freshwater transport |
|---|---|---|---|
| **Wainwright JAS SR** (df = 9) | **0.799**, 3-month avg (Feb-Apr) | **0.932**, 3-month avg (Feb-Apr) | **0.829**, 3-month avg (Feb-Apr) |
| **Peard Bay JAS SR** (df = 9) | Not significant | **-0.766**, 1-month avg (Jan) | *-0.683*, 4-month avg Jan-Apr |
| **Hope Basin Aug SR** (df = 4) | *-0.880*, 1-month avg (Feb) | *-0.866*, 4-month avg (Jun-Sep, prior year) | *-0.871*, 1-month avg (Feb) |
| **Hope Basin Sep SR** (df = 4) | **-0.962**, 11-month avg Sep prior year -Jul | **-0.994**, 10-month avg Nov prior year-Aug | **-0.991**, 12-month avg Aug prior year-Jul |

Correlations significant at p < 0.01 (bold); at *p<0.05 (italics)*.

**Table 4. Summary of bathymetry, gray whale prey and circulation drivers (transport and winds) used in iterative correlation analyses with Gray Whale (GW) Sighting Rates (SR) for three analytical areas in the northeastern Chukchi Sea.**

| Analytical Area | Bathymetry | Gray whale prey (see Table 2) | Transport @ A3 (see Table 3) | Winds over northern Chukchi Sea (see Figs 8, 9) |
|---|---|---|---|---|
| **Peard Bay** | Coastal shelf (20-40m deep) | Benthic species: amphipods cumaceans | **JAS GW SR** | **JAS GW SR** |
| | GW distributed along shore from Peard Bay to Point Barrow (Fig 4B) | | negatively correlated with average winter transports | **High 2009–2014** |
| | | | | Strong & persistent ENE winds in northern Chukchi in summer interrupt delivery of krill to head of Barrow Canyon, thereby making amphipods more accessible than krill |
| **Wainwright** | Offshore shelf | Benthic species: | **JAS GW SR** | **JAS GW SR** |
| | (40-50m deep) | amphipods, east flank of Hanna Shoal & near head of Barrow Canyon; increased abundance 2015–2017 (Fig 7) | positively correlated with average winter transports | **High 2015–2019** |
| | GW distribution clustered at the head of Barrow Canyon, NW of Wainwright (Fig 4B) | | | Weak & variable winds in the northern Chukchi in summer promote the convergent delivery of krill to the head of Barrow Canyon |
| | | Pelagic & epi-benthic species: | | |
| | | krill, possibly other spp. | | |
| **Hope Basin** | Inflow shelf (~40m deep) | Benthic species: amphipods, in trough 'hotspot' | **August GW SR** negatively correlated with winter or preceding summer transports | ICA results not significant |
| | GW distribution localized over 50-60m trough SW of Point Hope; southward extension along central channel in 2014, 2018, 2017 (Fig 5B) | Pelagic & epi-benthic species: krill, at frontal system coincident with trough slope (Bluhm et al. 2007) | **September GW SR** negatively correlated with long-term average transports | |

JAS = combined July-August-September SR.

Sightings of gray whales in the Bering Strait region during DBO cruises conducted from July-October suggest that the abrupt changes in calling activity may indicate a modest (*ca*. 30–40 km) shift in distribution away from the recorder locations, rather than a substantial change in gray whale phenology. While propagation modeling has not been conducted at this site, evidence from modeling studies elsewhere in the Bering Sea [47] suggests that gray whale calls are not likely detected beyond a radius of about 35km (20nm). So, for example, a shift of whales northward to the remaining amphipod beds in DBO2 or westward toward the DBO 3 hotspot near the IDL could take them out of detection range of the recorder. However, because the clear truncation in gray whale calling activity coincided with anomalous wintertime sea ice loss event in 2017–2019, this potential shift in phenology and/or distribution is noteworthy. Future deployments of recorders near the remnant amphipod beds in DBO2, or the benthic trough hotspot in DBO3 could help resolve some of the uncertainty regarding shifts in gray whale phenology in the Bering Strait region.

## Gray whale infaunal prey abundance, community composition, and environmental factors

Since 2010, the abundance of infaunal crustaceans that gray whales feed upon has significantly declined only in the DBO 2 region. This decline has been ongoing since the late 1980s [42, 43] leaving a reduced area of the northern Bering sea suitable for gray whale foraging. This change is concomitant with a shift in sediment quality due to faster currents, with the finer sediments required for tube building by *A. macrocephala* found only where current speed is reduced by land mass constriction south of Bering Strait [45]. Of note, the decline of infaunal crustaceans in DBO 2 is also coincident with an increase in polychaetes and bivalves [32, 33]. Collectively, these data suggest a physiological sensitivity of the dominant amphipods to warming

conditions in the DBO 2 region, with the potential to be outcompeted by organisms more tolerant of seawater changes and able to take advantage of increasing integrated and sediment chlorophyll carbon sources.

While there was no decline in infaunal crustacean abundance in DBO 3, a shift in dominant species was observed. Specifically, *Byblis* spp. (F. Ampeliscidae) was dominant from 1984–2004, but is now largely absent there, although it continues to be abundant in the DBO 5 region. The larger amphipod *Pontoporeia femorata* (F. Pontoporeidae) is now the dominant species in the northern portion of the DBO 3 hotspot, continuing to provide a benthic prey base for gray whales in the 'hotspot' area where they also appear to feed on advected krill [14]. Infaunal crustacean abundance in DBO regions 4 and 5 remained stable and without major changes in dominant fauna, at least through 2017.

## Environmental factors and pelagic-benthic coupling

What has influenced the changes or stability of the infaunal crustacean communities across the DBO 2–5 sites? On the face of it, the lack of change to infaunal crustacean abundance in DBO regions 3, 4 and 5 seems inconsistent with a fundamental tenant of the pelagic-benthic coupling model whereby loss of sea ice results in reduced organic carbon from ice-associated algae dropping to the sea floor and thereby cutting food supply for benthic macrofaunal communities [48]. Yet, our observations show that surface sediment chlorophyll increased as sea ice persistence decreased in DBO regions 2 and 4 (Table 2). Notably the significant decrease in sea ice persistence was associated with a significant increase in integrated water column chlorophyll only in DBO 2, although satellite observations indicate increasing productivity north of Bering Strait. These observations support the idea that changes in the timing and magnitude of water column primary production play an important role in carbon transfer to the seafloor [32]. Previous studies indicated that primary production is highest in the northern Bering Sea (DBO 1–2) in April/May, with a production peak north of Bering Strait (DBO 3) in June followed by peak production in the northeastern Chukchi Sea (DBO 4–5) in July [49]. Recently, however, pelagic primary production north of Bering Strait has increased in June and July [9, 50], which may be mismatched with the arrival of zooplankton grazers thereby leaving phytoplankton production to fall to the seafloor and feed benthic macrofauna. This effect on pelagic-benthic coupling may be especially true in the DBO 3 hotspot where deposition is greatest [51]. However, recent sediment trap results indicate that the early season breakup of sea ice at the DBO2-4 sites resulted in shorter periods of chlorophyll and diatom fluxes, thus potentially reducing the biological pump [52], emphasizing the importance of advective carbon to sustain northern DBO sites.

Clearly other environmental factors, such as sea ice thinning [10], increase in ocean heat and freshwater in the Chukchi Sea [5, 6], and changing sediment grain size associated with variability in current speed [32, 33, 45] are also influencing benthic macrofauna community structure. Specifically, increased water temperatures in DBO regions 2–4 is likely affecting the distribution of benthic species, as has been reported for the northeastern Atlantic where *Ampelisca* spp. have extended their distribution northward with warming seawater [53]. Of note, DBO5 stations in Barrow Canyon do not show increased bottom water temperatures, likely due to its dynamic nature and the seasonally variable water masses reported there [54]. Finally, increasing bottom water salinity since 2019 in DBO2 is perhaps indicative of a stronger Anadyr water signal at depth that may also influence nutrient availability to surface waters by mixing upward of AW after storm events. By comparison, DBO4 in the NE Chukchi Sea shows a decrease in bottom water salinity and a decline in integrated chlorophyll a, likely a response to reduced brine production with declining sea ice and variable nutrient content.

## Changes in gray whale distribution related to pelagic prey availability

Changes in gray whale distribution based on sightings were evident in the northeastern Chukchi Sea, and somewhat less so in the southern Chukchi Sea. Analyses of the ASAMM data show a mid-decade southwestward shift in distribution of roughly 110 km (60 nm), from coastal waters between Pt. Franklin and Pt. Barrow to waters offshore Wainwright. High gray whale sighting rates (SR) in the coastal Peard Bay area (2009–2014) were associated with strong and persistent easterly winds over the northern Beaufort-Chukchi region, while high SR in the Wainwright area (2015–2019) were associated with weak and variable winds that are conducive to krill remaining on the shelf over winter [28]. Winter-spring transports of volume, heat, and fresh water through Bering Strait were positively correlated with gray whale SR in the Wainwright area and negatively correlated with SR in the Peard Bay area (Table 4). This lagged relationship implies an advective timescale of ~4–6 months for biophysical signals to travel from Bering Strait to Barrow Canyon, which roughly comports with results from drifters traveling along that path in 90 days [55]. The heat signal, in particular, invites speculation that zooplankton development and survivability are enhanced in years with greater transport ultimately leading to improved summer foraging conditions near the head of Barrow Canyon. When krill remain on the northern Chukchi shelf, we postulate that they are subsequently aggregated via the convergence of northern Chukchi currents which promotes gray whale foraging near the head of Barrow Canyon. Although we have no confirmatory evidence (e.g., feces, stomach contents) that gray whales feed on krill in this area, they have been seen skim feeding on krill (Fig 1B) in association with foraging bowhead whales east of Pt. Barrow [28].

Changes in gray whale distribution in the southern Chukchi Sea were evaluated analytically using ASAMM data and, for waters south of 67˚ N, sightings during DBO cruises which provide qualitative information. Both data sources confirmed a strong and consistent aggregation of gray whales at a prey hotspot associated with a 50–60 m trough in the central Chukchi Sea. Furthermore, sightings during DBO cruises showed that this feeding hotspot extends across the international date line, with additional sighting of gray whales common along the Chukotka coast and associated with the western boundary of the central channel.

## Implications for the Gray Whale Unusual Mortality Event (UME)

In 2019, a gray whale UME was declared when the number of dead animals found stranded on beaches along their migration route between Alaska and Mexico increased roughly ten-fold [30]. A similar UME occurred in 1999–2000 and the emaciated condition of some of the dead whales during both events suggested starvation or nutritional deficit as a likely cause [30, 56]. Notably, ship strikes, entanglement in fishing gear, and predation by killer whales (*Orcinus orca*) contributed to the death toll during both events, while mortalities due to disease and biotoxins could not be clearly identified [57].

Whether or not gray whales were nutritionally stressed due to a change or reduction in prey availability related to environmental forcing, and/or to increased competition for food resulting from their burgeoning population size, remains a pivotal question with regard to both UME events [58]. The 1999–2000 UME closely followed the powerful 1997–1998 El Niño, while the onset of the 2019 UME was subsequent to extreme ocean heat events in the North Pacific and Gulf of Alaska from 2014–2016 [59], as well as the aforementioned ocean warming and wintertime sea ice loss that persisted in the northern Bering and Chukchi Seas from 2017–2019. The latter event was contributory to a mass mortality of seabirds [45] and likely influenced an ice seal UME [60].

The record on infaunal crustacean communities presented here shows that while gray whale benthic prey abundance declined in the northern Bering Sea from 2010–2019, it remained stable in the Chukchi Sea. Specifically, a significant decrease in amphipod abundance was observed in the DBO 2 region of the northern Bering Sea, a decline that extends a trend that began in the mid-1980s [45]. Conversely, there was no decrease in infaunal crustacean prey abundance evident in DBO regions north of Bering Strait, although there was a shift from a smaller (*Byblis* spp) to a larger (*Pontoporeia* spp) dominant amphipod species in the DBO 3 region. Meanwhile, iterative correlational analyses support the hypothesis that, due to changes in wind-driven circulation, gray whales may be finding more opportunities to feed on krill near the head of Barrow Canyon in the northeastern Chukchi Sea.

While changes in gray whale prey, either among benthic infauna or between benthic and pelagic species, will have nutritional ramifications, it seems unlikely that a change in diet alone would result in a dramatic upsurge in mortality. In other words, from the perspective of gray whale prey abundance, our observations provide context but not causation for poor nutritional condition of gray whales that feed north of Bering Strait. Conversely, if a portion of the gray whale population exhibits feeding-site-fidelity to the northern Bering Sea, that could result in nutritional stress for those animals. Indeed, there is evidence at the local scale that biophysical events that impact gray whale prey availability can lead to changes in phenology, a reduction in feeding, and signs of poor body condition. This was the case in 2005 when late spring ocean conditions off the Oregon coast markedly decreased abundance of epi-benthic mysids leaving the small population of gray whales that feed there over summer with a nutritional deficit [61]. A subsequent study showed body condition of whales in these whales responded to upwelling conditions along the Oregon coast, with animals in 'good' condition following strong upwelling years (2013–2015) and in comparatively 'poor' condition following years of weak upwelling (2016–2018) [62].

## The AMP model as a framework for future research

The Pacific Arctic marine ecosystem is in a state of transformation as it responds to amplified planetary warming [8]. In constructing the conceptual AMP model, we purposely imbedded mechanistic features such as transport through Bering Strait, pelagic-benthic coupling, advection, and upwelling as key drivers that influence biological outcomes in the region (Fig 2B) [29]. The challenge is to elucidate how these biophysical drivers interact in this productive inflow shelf system. Results from this and past studies suggests that this question may be best approached by investigating factors that (i) drive aspects of pelagic benthic coupling in the northern Bering and southern Chukchi Seas (DBO 2 and 3), and (ii) influence downstream advection, upwelling and ocean circulation dynamics in the northeastern Chukchi Sea (DBO 4 and 5). This type of approach was used to develop a mechanistic model for the persistence of benthic 'hotspots' in DBO regions 1–5, showing that those in regions 2, 3, and 5 rely on carbon supplied from upstream biological production, while those in regions 1 and 4 were supplied by local production [63]. As in past studies focused on macrofaunal biomass [32, 33], our observations of changes in infaunal crustacean prey abundance in DBO regions 2 and 3 suggest that the timing and magnitude of water column primary production, bottom water temperatures, bathymetric channeling, sediment grain size and the responses of other macrofaunal species may be as important as sea ice persistence in influencing pelagic-benthic coupling dynamics that lead to benthic community transitions.

The AMP conceptual model provides a rudimentary framework to guide future research aimed at elucidating changes in both benthic and pelagic realms of the Pacific Arctic. In the benthic realm, a focus on linking sea ice metrics [10] to biophysical data will provide a clearer

picture of their combined influence on productivity and community structure [33]. The addition of sediment trap data [52] can add additional detail to track pelagic-benthic coupling processes in the region. A key question for the Pacific Arctic region is: how important are krill as prey for upper trophic level predators? This requires investigating the more ephemeral prey aggregations in the epi-benthic and pelagic realm. The ICA results presented here suggests that the offshore shift of gray whale distribution after 2015 may have been influenced in part by the availability of krill in the northeastern Chukchi Sea. While this speculation builds on research focused on bowhead whale (*Balaena mysticetus*) foraging on krill in adjacent waters of the western Beaufort Sea [28], we have no direct evidence that gray whales are feeding on krill. Seabirds can also act as sentinels of krill availability, especially short tailed shearwaters (*Puffinus tenuirostris*) who feed on these advected prey in the northeaster Chukchi Sea more often now in the 1980s [64]. Using both marine mammal and bird species as ecosystem sentinels can alert researchers to changes in the pelagic realm across a broad range of space and time related to each species natural history [22] and we advocate for their continued inclusion in DBO and other ocean observatory protocols.

## Supporting information

**S1 Table. Summary of DBO cruises where a marine mammal watch was conducted.**
(PDF)

**S2 Table. Summary of survey effort, gray whale sighting and calculation of Sighting Rate (SR) from ASAMM 2009–2019.**
(PDF)

**S3 Table. Sampling details for crustacean abundance used in this paper.**
(PDF)

## Acknowledgments

We thank the captains and crews of the research vessels that provided safe operations and space for a marine mammal observer on the DBO research cruises. We appreciate the passive acoustic team from AFSC and CICOES who manually analyzed data from recorders on moorings in DBO2 and DBO3: Eric K. Braen, Stephanie L. Grassia, Eliza G. Ives, Daniel F. Woodrich, and Dana L. Wright. We thank the marine mammal observers, pilots, mechanics, programmers, flight followers, and organizations that enabled ASAMM surveys to be conducted safely and effectively, in particular Megan Ferguson, Amelia Brower, Amy Willoughby, NOAA Aircraft Operations Center, and Clearwater Air, Inc. The scientific results and conclusions, as well as any views or opinions expressed herein, are those of the author(s) and do not necessarily reflect those of NOAA or the Department of Commerce. Reference to trade names does not imply endorsement by the National Marine Fisheries Service or NOAA. JMG thanks past and current support staff at CBL for technical assistance, especially Stephanie Soques and Alicia Clark, and Alynne Bayard for GIS mapping and data analytics. Finally, we thank Dr. Karen Frey (Clark University) for providing Fig 2A, and Dr. Leigh Torres (OSU) and an anonymous reviewer for constructive comments that guided improvements to the final manuscript.

## Author Contributions

**Conceptualization:** Sue E. Moore, Janet T. Clarke, Stephen R. Okkonen, Jacqueline M. Grebmeier.

**Data curation:** Janet T. Clarke, Jacqueline M. Grebmeier, Catherine L. Berchok, Kathleen M. Stafford.

**Formal analysis:** Janet T. Clarke, Stephen R. Okkonen, Jacqueline M. Grebmeier, Catherine L. Berchok, Kathleen M. Stafford.

**Funding acquisition:** Jacqueline M. Grebmeier, Catherine L. Berchok, Kathleen M. Stafford.

**Project administration:** Sue E. Moore.

**Writing – original draft:** Sue E. Moore, Janet T. Clarke, Stephen R. Okkonen, Jacqueline M. Grebmeier.

**Writing – review & editing:** Sue E. Moore, Janet T. Clarke, Stephen R. Okkonen, Jacqueline M. Grebmeier, Catherine L. Berchok, Kathleen M. Stafford.

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
