## [Decision Letter · Decision Letter 0]

9 Nov 2021

PONE-D-21-20781

Changes in gray whale phenology and distribution related to prey variability and ocean biophysics in the northern Bering and eastern Chukchi seas

PLOS ONE

Dear Dr. Moore,

Thank you for submitting your manuscript to PLOS ONE. After careful consideration, we feel that it has merit but does not fully meet PLOS ONE’s publication criteria as it currently stands. Therefore, we invite you to submit a revised version of the manuscript that addresses the points raised during the review process.

As you will see from the Reviewer comments, major and mandatory revisions are required before this study may become acceptable for publication in PLOS ONE. In particular, the Reviewers raise concerns about some of the description of the methods, the statistical analyses, as well their interpretation and conclusions reached. Expanding on the link between gray whale distribution patterns and the physical mechanisms, especially as they pertain to physical variables shown, is recommended. The Reviewers also suggest improvements to the graphical representation of your results, including improved clarity of your figures, legends, and figure captions. The two Reviewers provide extensive feedback on how to address the concerns raised in their detailed comments below.

We look forward to receiving your revised manuscript.

Kind regards,

Caroline Ummenhofer

Academic Editor

PLOS ONE

Journal Requirements:

3. We note that Figure(s) 2A, 4A, 4B, 5, 7 and 9 in your submission contain map images which may be copyrighted. All PLOS content is published under the Creative Commons Attribution License (CC BY 4.0), which means that the manuscript, images, and Supporting Information files will be freely available online, and any third party is permitted to access, download, copy, distribute, and use these materials in any way, even commercially, with proper attribution. For these reasons, we cannot publish previously copyrighted maps or satellite images created using proprietary data, such as Google software (Google Maps, Street View, and Earth). For more information, see our copyright guidelines: http://journals.plos.org/plosone/s/licenses-and-copyright.

1. You may seek permission from the original copyright holder of Figure(s) 2A, 4A, 4B, 5, 7 and 9 to publish the content specifically under the CC BY 4.0 license.  

4. We note that Figure (1) in your submission contain copyrighted images. All PLOS content is published under the Creative Commons Attribution License (CC BY 4.0), which means that the manuscript, images, and Supporting Information files will be freely available online, and any third party is permitted to access, download, copy, distribute, and use these materials in any way, even commercially, with proper attribution. For more information, see our copyright guidelines: http://journals.plos.org/plosone/s/licenses-and-copyright.

1. You may seek permission from the original copyright holder of Figure (1) to publish the content specifically under the CC BY 4.0 license. 

Reviewers' comments:

Reviewer's Responses to Questions

**Comments to the Author**

1. Is the manuscript technically sound, and do the data support the conclusions?

Reviewer #1: Partly

Reviewer #2: Yes

2. Has the statistical analysis been performed appropriately and rigorously? 

Reviewer #1: No

Reviewer #2: Yes

3. Have the authors made all data underlying the findings in their manuscript fully available?

Reviewer #1: Yes

Reviewer #2: Yes

4. Is the manuscript presented in an intelligible fashion and written in standard English?

Reviewer #1: Yes

Reviewer #2: Yes

5. Review Comments to the Author

Reviewer #1: This manuscript compiles some excellent, longitudinal data on gray whale sightings, prey distribution and availability, and oceanography in the Arctic region that was collected synchronously. It is a very impressive dataset and the authors should be commended in their field efforts to collect these data. The questions addressed revolve around the changing oceanography in the Arctic region and impacts on prey availability and gray whale distribution over the sampling period. This research is very interesting, and answers are needed to better understand the impacts on such a large population of crustacean predators (gray whales) and the overall ecosystem response to rapid environmental change. In particular, I feel the changes in acoustic, oceanography and prey data over time and space are interesting and solid (Fig. 3, Fig 6, Table 2), but I feel the links to gray whale spatial distribution patterns are not well connected or presented.

While the data are impressive, I found the spatial analysis to be lacking in adequate description and standardization, which makes assessment of results and author interpretation challenging. In particular, the survey effort conducted during DBO cruises needs better description and evaluation. Presentation of raw numbers of sightings is misleading without standardization of this information by the amount of survey effort (i.e., km). Additionally, the authors discuss overlap between areas of high whale sightings and different prey availability, but this analysis is not described and results not displayed. This overlap analysis could be highly enlightening but needs more details and clarity. Furthermore, I am unclear how figure 5 was produced. It appears to be a GIS interpolation of crustacean sample data, but the input data is not described, nor the analysis methods. There are many types of interpolation methods, all of which have bias and assumptions, so its important to clearly describe the approach used.

Overall, I feel that more details on statistical methods are needed. Some results are presented that were not described in the methods section. Additionally, I think there is a need to link the research questions more clearly to the statistical analyses so that the work appears more directed and purposeful. Also, the authors split the data set temporally at 2014/15 or 2013/14 for different analyses and no justification is provided for selecting these dividing points, or explanation of why different times spans are analyzed for different data.

The ICA analysis seems like an interesting and robust method of time-series analysis but this is new to me so I cannot fully evaluate the approach or results. I did find the results hard to interpret so perhaps the results need more “hand-holding” for people like me who are unfamiliar with this method.

Discussion of gray whale foraging on krill, and the driving role of oceanography, is confusing to me as the authors do not present any data on krill collection or analysis, as far as I can tell (krill are not listed in Table 2). The reference to the Grebmeier manuscript in this same volume makes it hard to know what taxa were collected and included in this analysis. And the paper that is referenced regarding gray whale foraging on krill (Bluhm et al.) is in the south-central Chukchi Sea, which is not the same area that is discussed in this paper (DBO regions 4 and 5 in the northeastern Chukchi Sea). The Kim and Oliver paper do not mention euphausiids. I understand that the potential role of krill as a target species for foraging Arctic gray whales is emerging, so data is minimal, but I think it needs to be more clear that this new foraging strategy is still speculation due to the paucity of data.

The authors are clearly very knowledgeable about the ecology and oceanography of this region, and some ideas presented are very interesting, but I found the Discussion to be too long with a fair bit of speculation and assumptions based on ‘noteworthy’ ideas. Additionally, there were a few parts of the results and discussion sections that seemed redundant (same information presented or stated) so I think the Discussion can be trimmed down and written in a more concise and precise manner, with less speculation. I suggest the authors focus on the hard, supported conclusion their analysis reveals.

I found some of the figures hard to fully interpret due to a lack of full legends, description, and clarity. I think it’s also important to recognize that many readers (myself included) are not intimately familiar with the Artic region and the DBO project (sampling scheme, acronyms, etc.). So, any effort to improve labels and descriptions of methods would be helpful.

I provided many specific comments in the manuscript PDF regarding my questions and suggestions for improvements. I sincerely hope my feedback helps improve the analysis and presentation of the results so that the readers can see and understand the author’s interpretations and conclusions better. There is high value in this research, and I am excited to see the authors publish their study, but I feel that the analysis needs more attention and description.

Reviewer #2: This paper documented how the gray while phenology and distribution change seasonally and interannually, and provided the potential drivers, including hydrographic and biological properties, ice condition and wind forcing. I am particularly interested in the section where the authors investigated the impact of wind on the geological shift of the gray whale SR near the head of the Barrow Canyon. The authors suggested that gray whale distributes more in the Peard Bay area when persistent northeasterly wind blows, and more in the Wainwright area when the wind is weak and variable. I am curious the mechanism. The northeasterly wind can induce upwelling in the canyon which may in turn brings deep water onto the shelf, particularly in the coastal region. It seems to be consistent with the gray whale distribution in the Peard Bay which is confirmed in the coastal region (Fig. 7). By contrast, in the any other wind condition, the Alaskan Coastal Current flows off the shelf via the canyon, with a pumping-down process (Pickart et al., 2021). I suspect this might be the reason that in this wind regime, gray whale has a broader distribution in the Wainwright area where is still shallow and food-rich.

I understand the physical mechanisms are not the scope of the paper, but think it would be nice to briefly discuss these as the wind/hydrographic data have been presented. I also strongly recommend the authors to use high-res figures. Many figures are far from clear, e.g. the labels in Fig. 3, the years of the bar plots in Fig. 7, the labels on the maps of Fig. 8.

Over all, the paper is informative with such great datasets and improves our understanding of gray whale in the Bering and Chukchi Seas. I would thus recommend a minor revision.

The specific comments and questions:

Some place names are not shown, e.g. Hope Basin, Pt. Franklin, Pt. Barrow (be sure everybody knows that it is Utqiaġvik on the maps). The Wainwright and Peard Bay were first shown prior to the results, although they have been presented in Fig.7. I would suggest to add the names on the Fig.2.

Label the two panels in Fig.3 A and B. I guess the blue curves are the ice concentration in somewhere. Please make it clearer in the caption.

If I understand correctly, in Figure 5, the abundance map was interpolated using the data from the DBO and UTN stations shown in the figure. The authors need to remove the data away from the stations, which I think are fake and misleading.

Table 2 is out of the page, I cannot see the column of DBO5.

Line 336. remove the bracket in front of the equation.

What does the ellipse in Fig. 7 represent for?

The caption of Fig. 8 is not clear. What does the color in the maps of Fig. 8. Variance? If so, variance of what? Wind speed? Or any wind component? What are the blue contours? Is the mean atmospheric circulation actually the composite of the 10-m wind?

The summary in the Table 4 is very helpful!

6. PLOS authors have the option to publish the peer review history of their article (what does this mean?). If published, this will include your full peer review and any attached files.

Reviewer #1: **Yes: **Leigh Torres

Reviewer #2: No

---

## [Author Response · Author response to Decision Letter 0]

26 Jan 2022

I have responded to all comments in Decision Letter

I have revised the Cover Letter by adding the 'Role of Funder' statement, as requested

---

## [Decision Letter · Decision Letter 1]

24 Feb 2022

PONE-D-21-20781R1Changes in gray whale phenology and distribution related to prey variability and ocean biophysics in the northern Bering and eastern Chukchi seasPLOS ONE

Dear Dr. Moore,

Thank you for submitting your manuscript to PLOS ONE. After careful consideration, we feel that it has merit but does not fully meet PLOS ONE’s publication criteria as it currently stands. Therefore, we invite you to submit a revised version of the manuscript that addresses the points raised during the review process.

Both reviewers suggest acceptance of your manuscript for publication with PLOS One and I concur with their assessment.  Some final suggestions for edits are included in the detailed comments by the reviewer below to help with the clarity of the study overall and hence the manuscript is returned for minor revisions.

We look forward to receiving your revised manuscript.

Kind regards,

Caroline Ummenhofer

Academic Editor

PLOS ONE

Journal Requirements:

Reviewers' comments:

Reviewer's Responses to Questions

**Comments to the Author**

1. If the authors have adequately addressed your comments raised in a previous round of review and you feel that this manuscript is now acceptable for publication, you may indicate that here to bypass the “Comments to the Author” section, enter your conflict of interest statement in the “Confidential to Editor” section, and submit your "Accept" recommendation.

Reviewer #1: All comments have been addressed

Reviewer #2: All comments have been addressed

2. Is the manuscript technically sound, and do the data support the conclusions?

Reviewer #1: Yes

Reviewer #2: Yes

3. Has the statistical analysis been performed appropriately and rigorously? 

Reviewer #1: Yes

Reviewer #2: Yes

4. Have the authors made all data underlying the findings in their manuscript fully available?

Reviewer #1: Yes

Reviewer #2: Yes

5. Is the manuscript presented in an intelligible fashion and written in standard English?

Reviewer #1: Yes

Reviewer #2: Yes

6. Review Comments to the Author

Reviewer #1: I appreciate the revision of this manuscript. I could follow how the data were used and applied to answer different questions much better than previously. I have some remaining comments below that I think the authors should address, but these should not be onerous and will help with clarity. I think this paper does a nice job of bringing together many datasets in a biologically complex region undergoing environmental change to reveal some patterns, highlight some unexpected findings, and pose interesting hypotheses worth follow up research. Great work. [Of note, I could never download or view Table 2. This was not included in the PDF provided for review. It is highly referenced in the text, so its unfortunate I could not review it.]

Fig 2 legend: Describe what the green and white lines are. Which are describing what?

Line 157: Change “distribution” to “presence”, since distribution includes understanding where animals are not and this assessment has not been conducted. “…sightings from DBO cruises provide data on gray whale presence,…”. I appreciate the clarification on the use (and non-use) of these data.

Line 171: You say three geographic areas but only 2 are listed. I am confused.

Lin 200-201: Change “Spatial analysis” to “spatial interpolation” in this sentence: “Spatial interpolation was accomplished using geographical information system software.” The data were not actually analyzed using the IDW, but rather interpolated for visual assessment. The Spearman’s rho rank correlation did the analysis part. I do appreciate the added details on the IDW method. Thanks.

Thank you for the improved description of what the ICA does and tests.

Visual detections section – This is much improved to clarify interpretation and limitations of data. For Figure 4, can you add the DBO areas? You refer to these areas in the text (lines 257-259) so I found myself awkwardly bouncing between the text, Figure 2A and Figure 4A to understand what was described. Addition of the DBO boxes would also help understand where data was collected and where it was not, since this is mainly presence-only data.

Line 275-277: I just don’t see this shift clearly in the data. Maybe because of overlapping data points. I suggest making the points more transparent, or using a kernel density approach to describe these shifted distribution patterns.

Line 277-279: Are you refereeing to the west of Hope Point? Can you specify this? Otherwise I am not sure where the “trough” is.

Line 281: Can you remind the reader of where these data were derived from? “with reference to observed changes in infaunal crustacean prey abundance and community composition derived from…”.

Figure 5: resolution of this figure is still low so I can’t see the “closed gray circles” well. Also, I am unclear about the source of the “all stations”. Are these from the Arctic Data Center? Can you clarify?

Figure 6: Please describe the red line for DBO 2. Is it for the open or closed circles? Or both?

Figure 7: This is a handy figure. Thanks for adding. But can you describe how these abundance time series visualizations were generated? I made have read it in the methods, but forgot by the time this figure came up. I think if you just add it to the legend that would help me.

Can you plot Hanna Shoal on Figure 5?

I think you should mention the scale of Fig 5? That it is a decadal view of crustacean abundance. As Fig 6 shows, this abundance is dynamic, so the scale of the data representation should be made clear.

Line 339: Where do these observations of “feeding gray whales” come from? ASAMM? I think it would be good to describe.

Line 437-438: Did you conduct a ICA analysis to determine this negative result of “no clear relationship”? This should be explained and detail. It strengthens the results show in Fig 9.

Does the ICA account for interactions between the predictor variables? I would guess that volume, heat and freshwater transport interact with each other (likely in a non-linear way). Can you add a description into the methods about if/how ICA deals with those interactions?

Line 497-498: The percentages add up to more than 100%. So what do they represent?

Line 529: Can you remind us when these DBO cruises were? Because the point here is that sightings of gray whales continued to be made during the DBO cruises even when the acoustic detections dropped. So, adding the overlapping time frame component to this sentence would be helpful.

Thanks for the acknowledgement. 

Reviewer #2: (No Response)

7. PLOS authors have the option to publish the peer review history of their article (what does this mean?). If published, this will include your full peer review and any attached files.

Reviewer #1: **Yes: **Leigh Torres

Reviewer #2: No

---

## [Author Response · Author response to Decision Letter 1]

7 Mar 2022

A Response to Reviewer for R2 file has been uploaded

---

## [Editor Report · Decision Letter 2]

11 Mar 2022

Changes in gray whale phenology and distribution related to prey variability and ocean biophysics in the northern Bering and eastern Chukchi seas

PONE-D-21-20781R2

Dear Dr. Moore,

We’re pleased to inform you that your manuscript has been judged scientifically suitable for publication and will be formally accepted for publication once it meets all outstanding technical requirements.

Kind regards,

Caroline Ummenhofer

Academic Editor

PLOS ONE
---

## [Editor Report · Acceptance letter]

23 Mar 2022

PONE-D-21-20781R2 

Changes in gray whale phenology and distribution related to prey variability and ocean biophysics in the northern Bering and eastern Chukchi seas 

Dear Dr. Moore:

I'm pleased to inform you that your manuscript has been deemed suitable for publication in PLOS ONE. Congratulations! Your manuscript is now with our production department. 

Kind regards, 

on behalf of

Dr. Caroline Ummenhofer 

Academic Editor

PLOS ONE